# The Dipole Resonator and Dipole Waveguide Insulator in Dense Liquid Medium

## Mikhail Mironov

Theoretical Department, Andreyev Acoustics Institute, 117449 Moscow, Russia; mironov_ma@mail.ru

**Abstract:** In this paper, the propagation of sound in an acoustically narrow waveguide, the wall of which is lined with identical dipole resonators and masses on springs, is theoretically considered. It is shown that, in the frequency range above the resonant frequency of the resonators, sound waves exponentially attenuate, and the waveguide is locked. The width of this range depends on two parameters—the ratio of the cross-sectional areas of the resonators and the waveguide and the ratio of the mass of the resonator to the mass of the medium displaced by it. As the resonator mass decreases, the locking band width expands and may become infinite.

**Keywords:** resonator; dipole; waveguide; sound locking

## 1. Introduction

The Helmholtz resonator (HR) is an effective sound scatterer and absorber at its resonant frequency. An HR attached to a waveguide wall fully reflects the sound wave with a frequency equal to the resonant frequency of the resonator, and the waveguide turns out to be locked. When lining the wall of an acoustic waveguide with identical HRs (Figure 1), the effect of waveguide isolation occurs: in the frequency range lying above the resonant frequency of the resonators and below a certain frequency, the waves do not propagate, but exponentially attenuate. The locking band of the waveguide turns out to be significantly wider than the locking band for the resonator. This effect was investigated for the first time in [1–3] in the 1970s, and the construction, which consisted of a set of identical resonators was named as a waveguide insulator (WI). The explanation of this effect is as follows. At frequencies above the resonant frequency, the resonator throats have a mass-type impedance. The medium in the throat moves in the antiphase with the applied pressure. Accordingly, at a positive pressure, in the throat of the resonator, the medium moves inside the waveguide. This corresponds effectively to negative compressibility. In addition to the positive compressibility of the medium in the waveguide, the total effective compressibility may turn out to be negative. Consequently, the wave propagation velocity will be imaginary and the wave will be exponentially attenuated. Currently, media with resonant inclusions are widely studied as part of the development of "acoustic metamaterials", and the number of articles on this topic is estimated to be in the tens, if not hundreds. Some examples of this issue are discussed below.

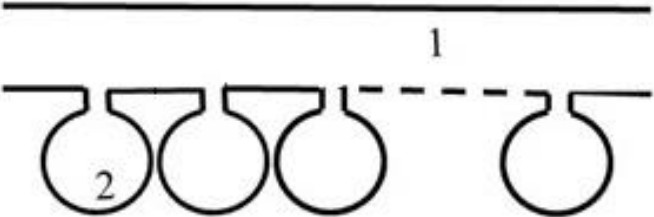

**Figure 1.** Waveguide 1 with attached Helmholtz resonators 2.

In [4], a general theoretical framework for the description of sound wave interaction with the resonant metasurface was presented. This approach is independent of the nature of resonators and the excitation. The equations governing the behaviour of the metasurface are upscaled from the rigorous description of its unit cell using a two-scale asymptotic homogenization. The model is capable of describing sound interaction with the array of resonators positioned above or on the substrate, so that the out-of-plane direction becomes an additional degree of freedom in the design. It was shown that the behaviour of the resonant surface is described in terms of its effective admittance. This theory was validated by experiments performed in the impedance tube and in the anechoic environment using a surface array of HRs.

In [5], a homogenization method based on two-scale matched asymptotic expansion techniques for arrays of HRs with two opposite-oriented throats was also used. In the resulting effective model, the array was replaced by a homogeneous and anisotropic medium accounting for the cavities of the resonators, while jump conditions applied across a fictitious interface accounting for the necks of the resonators. The model is able to accurately describe resonators open with necks at both extremities (termed two-sided) or open at a single extremity (termed one-sided).

A broadening of the noise attenuation frequency band by periodic HR arrays due to the coupling of the Bragg reflection and HRs' resonance was investigated in [6]. The study provides a method for noise control applications of ventilation ductwork systems by utilizing the advantage of periodicity to broaden the noise attenuation band and considering the insufficient duct length for a pure periodic HR array. It has a potential application in actual noise control with a limitation of the available completed duct length for HRs' installation.

A tonraum resonator, one type of HR, incorporated into the design of a plate-type acoustic metamaterial was studied in [7]. This study presented a proof-of-concept design of a membrane-type acoustic metamaterial without the need for membrane pretension and platelet(s). Its main aim was to extend the design to a large-scale set-up for potential applications in the industry, especially in the cabins of transport systems.

The study of [8] was devoted to the experimental and numerical verification of the WI theory [1–3]. The investigated model of a waveguide insulator (WI) was a small-diameter tube with three identical HRs attached to the tube wall. The chambers of each resonator were connected with the tube by means of six holes in the tube wall. The theoretical resonant frequency of resonators was equal to 3.3 kHz. Three findings from the paper should be mentioned. (i) The WI is effective in a wide frequency range. The maximum efficiency is observed at frequencies close to the resonant frequency and is 35–40 dB. The actual locking frequency range of the WI (more than 15 dB reduction) is observed at the frequency band from 3.3 kHz to 6.2 kHz. (ii) The results of numerical and analytical calculations agree with the results of the experiment. (iii) The WI theory is essentially based on the method of homogenization - the replacement of discrete inhomogeneities in the waveguide wall with continuous "distributed" wall impedance. The experiment and numerical calculation showed that homogenization was applicable even with a small number of resonators, in this case three.

A metaliner, whose unit cell consists of two different HRs, was proposed in [9]. In order to predict the sound insulation and absorption of a metaliner in a duct with a flow, an effective impedance model of the metaliner, including the effect of a flow, was established. Experiments showed that designed metaliners exhibit a transmission loss above 45 dB/m within 65% of the target frequencies of 500 and 1000 Hz for different flow speeds of 0, 17, and 34 m/s.

HRs are monopoles, which are sources of volumetric velocity. In [10], dipole resonators—masses on a spring, which are both sources of volumetric force, are considered. A dipole resonator, just like an HR, locks a narrow waveguide at its resonant frequency.

A sound absorber, as a combination of monopole and dipole resonators in a narrow waveguide, was considered in [11]. The optimum parameters of the resonators that provide the maximum absorption of acoustic power are determined. The results of an experimental

study of a two-resonator absorbing system were presented. A 95% of acoustic power absorption was achieved.

The locking of sound by a dipole resonator installed at the exit from a narrow pipe was theoretically and experimentally studied in [12]. The efficiencies of locking by dipole and monopole resonators were compared, and the dipole resonator was found to be more efficient than the monopole resonator.

The results of an active dipole investigation were given in [13,14]. The active resonator is modelled as a sound source with acoustoelectric feedback, which allows the correction of its motion depending on the sound field measured in the vicinity of the resonator. The open-end noise reduction of about 15–30 dB by means of dipole resonator was experimentally demonstrated. The active broadening of the frequency band of the dipole resonator was proposed and experimentally checked. A trial active dipole reflector effectively reduced noise in the band from 450 Hz to 900 Hz.

The scattering of a plain sound wave by an array of small monopole–dipole scatterers (HRs vibrating on springs) was theoretically considered in [15]. It was shown that, at a certain friction in the resonators, an array, whose spatial periods did not exceed the half-wavelength of sound, served as an efficient absorber for resonance-frequency sound.

Recent progress in acoustic metamaterial science were outlined in the review of [16], where the basic classification, underlying physical mechanism, application scenarios, and emerging research trends for both passive and active noise-reduction metamaterials were presented.

To the best of the author's knowledge, the lining of an acoustic waveguide wall with dipole resonators has not yet been considered in the literature. The main goal of the present paper is to achieve a similar effect to WI for a set of identical dipole resonators. It was shown that the placement of a set of identical dipole resonators on the walls of the waveguide led to a change in the effective density of the medium in the waveguide. In the frequency band above the resonant frequency of the resonators, the effective density turned out to be negative. Accordingly, the square of the effective wave propagation velocity, as well as when using monopole resonators, turned out to be negative, which led to locking of the waveguide.

The paper is organized as follows. In Section 1, a generalization of the theory of dipole resonator [10] is given, taking into account the buoyant force acting on an oscillating mass in a sound field. It is shown that, for a dense medium (water), this additional force can noticeably expand the locking band in the waveguide. Section 2 presents the theory of WI for a chain of dipole resonators and some examples of calculated decay decrement. The Conclusion section summarizes the main results obtained in this study.

## 2. A Single Dipole Resonator in a Waveguide

Here, we consider a generalization of the theoretical part of this study [10], which consists of taking into account the buoyant force acting on the mass of the dipole resonator. The initial model of the dipole resonator [10], also implemented in experiments [10–14], is shown in Figure 2a. The cylinder 2, filled with the same medium as the waveguide 1, is tightened on one side by a rubber membrane 3 and is open on another side. The membrane acts as a spring but also has an effective moving mass. The mass of the medium 4, which fills the cylinder, is the attached mass. Figure 2b shows a generalized model of a dipole resonator. It differs from the original model in that the mass of the dipole 2 has a finite volume. The calculation model of the dipole resonator is shown in Figure 2b.

In the waveguide 1 there is a mass 2 suspended on springs 3. Together with the mass 2, the area of the medium 4 (the attached mass of the dipole) oscillates. The equation of motion of a dipole resonator has the form (see, e.g., [17], ch.1, p. 53, (11.9)):

$$m\dot{v} = -\mu(\dot{v} - \dot{u}) - \kappa \int v\,dt + \rho V_0 \dot{u} \qquad (1)$$

$m$ is the mass of the dipole resonator 2, $\mu$ is the attached mass 4 (generally speaking, a complex quantity, whose imaginary part takes into account friction), $\kappa$ is the stiffness of the spring 3 (can also be a complex quantity), $\rho$ is the density of the medium, $V_0$ is the volume of the oscillating mass 2, $v$ is the vibrational velocity of the resonator mass, and $u$ is the specified vibrational velocity of the medium outside the resonator. The last term in the right part (1) (buoyant force), which is essential for the dense (aquatic) medium, was not taken into account in [10]. A force from the resonator acting on the medium is equal to:

$$F_0 = +\mu(\dot{v} - \dot{u}) - \rho V_0 \dot{u}$$

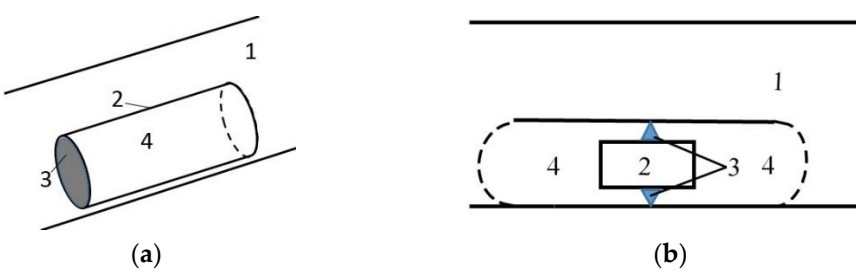

**Figure 2.** (**a**) Initial model of the dipole resonator [10]: 1—waveguide; 2—cylinder; 3—tightened membrane; 4—medium inside cylinder. (**b**) Calculation model of a dipole resonator in a waveguide: 1—waveguide; 2—the mass of the dipole; 3—the springs of the dipole; 4—the medium filling the dipole—its attached mass.

Considering that the last term in this formula is the force at which the element of the medium replaced by the resonator would act upon, the additional force added by the resonator is equal to:

$$F = +\mu(\dot{v} - \dot{u}) \tag{2}$$

The elastic force is not included in this force since it acts on an external support: on the wall of the waveguide.

Passing to the spectral representation, expressing $v$ by $u$ and substituting it in (2), we obtain:

$$F = \mu(-i\omega)\frac{\omega^2(m - \rho V_0) - \kappa}{-\omega^2(m + \mu) + \kappa}u \tag{3}$$

Next, wave the transmission coefficient $W_d$, through the section of the waveguide in which the dipole resonator is installed, is calculated. This problem is reduced to a boundary value problem with boundary conditions at this stage: the vibrational velocity of the acoustic field is continuous, and the pressure experiences a jump equal to the force (3) divided by the cross-sectional area of the waveguide. Omitting elementary intermediate calculations, we obtain a final answer:

$$W_d = \cfrac{1}{1 + \cfrac{i\omega\mu}{2\rho c S} \cdot \cfrac{1 - \left(\cfrac{\omega}{\omega_{d0}}\right)^2 \cfrac{m - \rho V_0}{m + \mu}}{\left(\cfrac{\omega}{\omega_{d0}}\right)^2 - 1}} \tag{4}$$

where $\omega_{d0} = (\kappa/(m + \mu))^{1/2}$ -, the resonant frequency of the dipole resonator $c$ is the speed of sound in the medium. This formula differs from Formula (4) of [10] by the presence of the term $-\rho V_0$, the mass of the displaced medium. If the mass of the dipole is equal to the mass of the displaced medium $-m = \rho V_0$, then Formula (5) exactly coincides with the formula in [10].

At the resonant frequency of the dipole resonator, the transmission coefficient is equal to zero, regardless of the value $-\rho V_0$. The behaviour of the transmission coefficient

near the resonant frequency is quantitatively convenient for describing the width of the frequency band $\Delta\omega/\omega_0$, which the transmission coefficient does not exceed, using the modulus that is a given value $\varepsilon$. The calculation of the so-defined band width gives the following expression:

$$\frac{\Delta\omega}{\omega_0} = \frac{1}{2}\frac{\omega_0\mu}{\rho cS}\cdot\frac{\rho V_0 + \mu}{m + \mu}\varepsilon \tag{5}$$

Let us analyse this formula in relation to two constructions of dipole resonator. The first design is a ball on a spring. The density of the ball material is equal to $\rho_0$, the radius of the ball is equal to $R_0$. Given, that the attached mass of the ball is equal to half the mass of the displaced medium, Formula (5) is rewritten as:

$$\frac{\Delta\omega}{\omega_0} = \frac{1}{3}\frac{\omega_0}{c}\frac{R_0{}^3}{S}\cdot\frac{2}{\frac{4}{3}(\rho_0/\rho) + 2/3}\varepsilon \tag{6}$$

If the density of the ball material is equal to the density of the medium $\rho_0 = \rho$, the band width is equal to $\frac{\Delta\omega}{\omega_0} = \frac{1}{3}\frac{\omega_0}{c}\frac{R_0{}^3}{S}\varepsilon$. If the density of the ball material is zero, the band width is tripled compared to the previous case: $\frac{\Delta\omega}{\omega_0} = \frac{1}{3}\frac{\omega_0}{c}\frac{R_0{}^3}{S}3\varepsilon$.

The second design, schematically shown in Figure 2b, is a segment of a cylinder with length $L$ and radius $R$, inside which a cylindrical piston with length $L_0$ and the same radius as the cylinder is suspended on springs. The masses included in Formula (5) are equal to:

$$\rho V_0 = \rho L_0\pi R^2, \ \mu = \rho(L - L_0)\pi R^2 + \rho\alpha\pi R^3, \ m = \rho_0 L_0\pi R^2 \tag{7}$$

Here, the coefficient for the terminal correction of the attached mass $\alpha \sim 1$ is introduced. Substituting (7) into (5), we obtain the following expression for the band width:

$$\frac{\Delta\omega}{\omega_0} = \frac{1}{2}\frac{\omega_0[(1 - L_0/L) + \alpha R/L]}{cS}\pi LR^2\cdot\frac{1 + \alpha R/L}{(\rho_0/\rho)(L_0/L) + (1 - L_0/L) + \alpha R/L}\varepsilon \tag{8}$$

Let us consider a few limiting cases.
(1) $L_0 = 0$ the length of the oscillating mass is equal to zero:

$$\frac{\Delta\omega}{\omega_0} = \frac{1}{2}\frac{\omega_0[1 + \alpha R/L]}{cS}\pi LR^2\cdot\varepsilon$$

The band width is proportional to the sum of the volume of the cylinder and the attached volume of the end sections.
(2) $L_0 = L$—the length of the oscillating mass coincides with the length of the entire dipole:

$$\frac{\Delta\omega}{\omega_0} = \frac{1}{2}\frac{\omega_0[\alpha R/L]}{cS}\pi LR^2\cdot\frac{1 + \alpha R/L}{(\rho_0/\rho) + \alpha R/L}\varepsilon$$

(2a) $\rho_0 = \rho$—the density of the oscillating mass is equal to the density of the medium:

$$\frac{\Delta\omega}{\omega_0} = \frac{1}{2}\frac{\omega_0\alpha}{cS}\pi LR^3\cdot\varepsilon$$

The width of the frequency band is proportional to the end correction of the end sections.
(2b) $\rho_0 = 0$—the density of the oscillating mass is equal to zero:

$$\frac{\Delta\omega}{\omega_0} = \frac{1}{2}\frac{\omega_0[1 + \alpha R/L]}{cS}\pi LR^2\cdot\varepsilon$$

The width of the frequency band is proportional to the sum of the volume of the cylinder and the attached volume of the end sections, the same as case 1.

### 3. Dipole Waveguide Insulator

A dipole waveguide insulator is a wall-mounted dipole resonator with a surface density of $n$. Resonators oscillate under the influence of a sound field in the medium and have a reverse effect on it in the form of a force acting on the medium. The force acting on the medium from resonators with a surface density $n$ is obtained from Formula (3) by multiplying it by $n$:

$$f = \mu(-i\omega)n\frac{\omega^2(m - \rho V_0) - \kappa}{-\omega^2(m + \mu) + \kappa}u \tag{9}$$

We substitute this force into the Euler equation as an additional term:

$$\varrho(-i\omega)u = -\frac{dp}{dx} + \mu(-i\omega)\frac{n \cdot l}{S}\frac{\omega^2(m - \rho V_0) - \kappa}{-\omega^2(m + \mu) + \kappa}u$$

Here, $S$ is the cross-sectional area of the waveguide, and $l$ is its perimeter. Furthermore, the second term on the right part is transferred to the left part:

$$\left(\varrho - \mu\frac{n \cdot l}{S}\frac{\omega^2(m - \rho V_0) - \kappa}{-\omega^2(m + \mu) + \kappa}\right)(-i\omega)u = -\frac{dp}{dx} \tag{10}$$

The form of the Euler Equation (10) allows one to enter the effective density of the medium:

$$\rho_{ef} = \varrho - \mu \cdot \frac{n \cdot l}{S} \cdot \frac{\omega^2(m - \rho V_0) - \kappa}{-\omega^2(m + \mu) + \kappa} = \varrho\left(1 - \frac{\mu}{\varrho} \cdot \frac{n \cdot l}{S} \cdot \frac{\omega^2(m - \rho V_0) - \kappa}{-\omega^2(m + \mu) + \kappa}\right) \tag{11}$$

Let us introduce a dimensionless parameter characterizing the ratio of the linear attached mass of the resonators to the linear mass of the medium in the waveguide:

$$\sigma = \frac{\mu}{\varrho} \cdot \frac{n \cdot l}{S} \tag{12}$$

The expression for the effective density is rewritten as:

$$\rho_{ef} = \varrho\left(1 + \sigma\frac{\omega^2(m - \rho V_0) - \kappa}{\omega^2(m + \mu) - \kappa}\right) = \varrho\frac{\omega^2[(m + \mu) + \sigma(m - \rho V_0)] - \kappa(1 + \sigma)}{\omega^2(m + \mu) - \kappa} \tag{13}$$

It follows from (13) that in the frequency range:

$$\omega_1 < \omega < \omega_2 \tag{14}$$

where

$$\omega_1 = \sqrt{\kappa/(m + \mu)}, \quad \omega_2 = \omega_1\sqrt{(1 + \sigma)/[1 + \sigma \cdot \frac{m - \rho V_0}{m + \mu}]} \tag{15}$$

the effective density is negative. Let us suppose that the isentropic compressibility of the medium in the waveguide and inside the resonators $\beta$ does not change. This supposition is equivalent to the initial statement about the pure dipole-type action of the resonators (no additional volume velocities). Then, the square of the effective speed of sound

$$c_{ef}^2 = \frac{1}{\rho_{ef}\beta}, \tag{16}$$

in this frequency range is also negative; a locking of the wave occurs, similar to the effect of waveguide isolation, when using Helmholtz resonators [1–3].

It follows from (14) and (15) that the width of the locking band:

$$\Delta = \frac{\omega_2}{\omega_1} = \sqrt{\frac{(1+\sigma)(m+\mu)}{m(1+\sigma)+\mu-\sigma\rho V_0}} \qquad (17)$$

is maximal at $m = 0$ and is equal to:

$$\Delta_{max} = \sqrt{(1+\sigma)\frac{\mu}{\mu-\sigma\rho V_0}} \qquad (18)$$

In the case of the inequality:

$$m(1+\sigma)+\mu-\sigma\rho V_0 < 0 \qquad (19)$$

the locking band width is not limited above, since the effective density at frequencies above the resonant frequency of the dipole remains negative at all frequencies. In this case, in the limit $\omega \to \infty$, the effective density tends towards a negative value independent of the frequency:

$$\rho_{ef\infty} = -\varrho\frac{\sigma\rho V_0 - m(1+\sigma)-\mu}{(m+\mu)} \qquad (20)$$

In particular, if the resonator mass is equal to zero, the maximum effective density taking into account (12) is:

$$\frac{\rho_{ef\infty}}{\varrho} = -\frac{\sigma\rho V_0 - \mu}{\mu} = -\frac{nl}{S}V_0 + 1 \qquad (21)$$

Let us consider the lining of a waveguide with dipole resonators as an example, as shown in Figure 3.

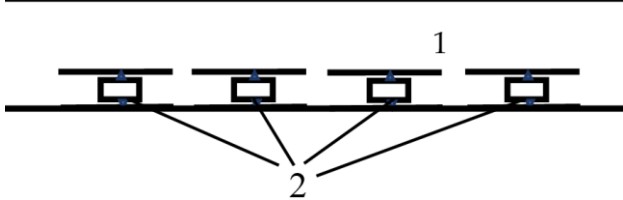

**Figure 3.** Lining of the waveguide 1 with dipole resonators 2.

We rewrite Formula (13) in dimensionless form:

$$\widetilde{\varrho}_{ef} = \frac{\varrho_{ef}}{\rho} = \frac{\Omega^2\left(1+\frac{\sigma(\widetilde{m}-1)}{\widetilde{m}+\widetilde{\mu}}\right)-1-\sigma}{\Omega^2-1} \qquad (22)$$

This formula includes four dimensionless parameters:

$\Omega = \omega/\omega_1$—the sound frequency normalized to the resonant frequency of the isolated dipole;
$\widetilde{m} = m/(\rho V_0)$—the normalized mass of the dipole;
$\widetilde{\mu} = \mu/(\rho V_0)$—the normalized attached mass of the dipole;
$\sigma = \widetilde{\mu}\cdot V_0 nl/S$—the filling factor of the waveguide surface with resonators.

In Figure 4, the frequency dependencies of the real part of the effective density for various values of the normalized mass $\widetilde{m}$ and fixed dimensionless parameters $\widetilde{\mu} = 0.2$, $\sigma = 0.4$ are given. A small absorption was introduced into the calculation formula by replacing $\Omega^2 \to \Omega^2(1+i\gamma)$, $\gamma = 0.1$ in (22). The main focus is on the frequency domain, in which the effective density is negative. Here, the phase velocity and the wave number $\xi_{ef} = \omega/c_{ef}$ turn out to be purely imaginary. Accordingly, the wave turns out to be

exponentially damped. The effective wave number is expressed in terms of the effective density by the formula (see (16)):

$$\xi_{ef} = \omega/c_{ef} = \omega\sqrt{\varrho_{ef}\beta} = \omega\sqrt{\frac{\varrho_{ef}}{\rho}\cdot\rho\beta} = \frac{\omega}{c}\sqrt{\tilde{\rho}} \tag{23}$$

Figure 5 shows the normalized imaginary parts of the wave numbers, calculated according to the formula:

$$\tilde{\xi} = \frac{\xi_{ef}}{\omega/c} = \sqrt{\tilde{\rho}}$$

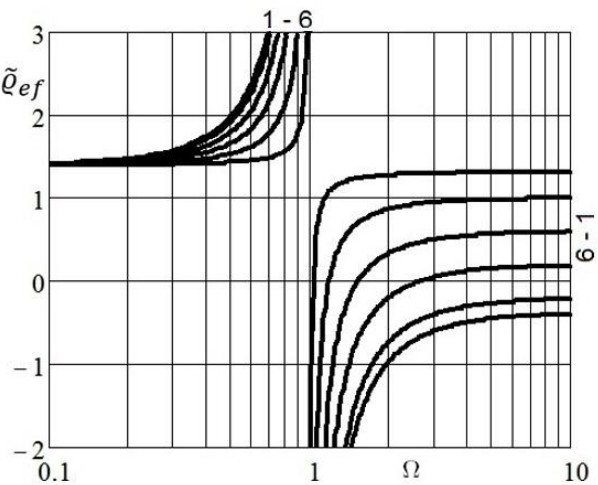

**Figure 4.** The dependence of the effective density $\tilde{\varrho}_{ef}$ on the frequency $\Omega$ at different values of the parameter: $\tilde{m}: 1 - \tilde{m} = 2;\ 2 - \tilde{m} = 1;\ 3 - \tilde{m} = 0.4;\ 4 - \tilde{m} = 0.2; 5 - \tilde{m} = 0.1; 6 - \tilde{m} = 0.07.$

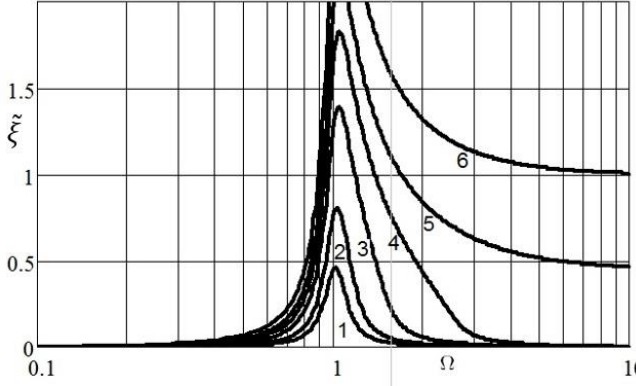

**Figure 5.** The normalized imaginary part of the wave number $\tilde{\xi}$ at different values of the parameter: $\tilde{m}: 1 - \tilde{m} = 2;\ 2 - \tilde{m} = 1;\ 3 - \tilde{m} = 0.4;\ 4 - \tilde{m} = 0.2;\ 5 - \tilde{m} = 0.1;\ 6 - \tilde{m} = 0.07.$

With a decrease in the mass of the dipole, the frequency band in which the wave exponentially attenuates increases, and with a mass equal to $\tilde{m} = 0.1$ (curve 5 in Figure 5), it becomes infinite. Physically, a decrease in the mass of the dipole compared to the mass of the displaced medium is possible for liquid media (water). In a gaseous medium, the best effect is obtained when the physical mass of the dipole is completely eliminated by reducing the volume $V_0$. Assuming in (13) $\tilde{m} = 0$, $V_0 = 0$, we obtained an expression for the locking band width for the gas medium:

$$\Delta_{gas} = \sqrt{1+\sigma} = \sqrt{1 + \frac{\mu}{\varrho}\frac{n\cdot l}{S}} \tag{24}$$

In accordance with this formula, the larger the fraction occupied by resonators in the waveguide, the wider the locking band.

## 4. Conclusions

In this paper, the lining of a waveguide with dipole resonators is theoretically considered. Similar to using monopole resonators (HRs), dipole resonators create the effect of locking sound in the frequency range that lies above the resonant frequency of the resonators. The physical interpretation of this effect is as follows. At frequencies above the resonant frequency, the force acting on the medium from the resonator (3) is in phase with the vibrational acceleration of the medium $-i\omega u$ and is proportional to it. The resonator pushes the medium, creating additional acceleration. The action of many resonators is equivalent to the addition of negative density. This negative addition in the frequency range (14) and (15) makes the total density negative, and the phase velocity of propagation purely imaginary. As a consequence, the wave number in this frequency range is imaginary, which corresponds to an exponential attenuation of sound. The buoyant force, acting on the mass of the dipole resonator, reveals a new, previously unknown result. It was shown that the buoyant force expanded the locking band and, under certain conditions imposed on the mass, its volume, the filling density of the waveguide, and the width of the locking band (17) are unlimited from above. At the limit of high frequencies, the effective density turned out to be constant and negative, the phase velocity of the wave is constant and imaginary.

The development of a specific design of a dipole waveguide insulator is beyond the scope of this work. A simple physical model that can be used for laboratory experiments may consist of a set of resonators in the form of cylindrical tubes, with one end closed by a stretched membrane and the other end open (see Figure 2a).

**Funding:** This research received no external funding.

**Institutional Review Board Statement:** Not applicable.

**Informed Consent Statement:** Not applicable.

**Data Availability Statement:** Not applicable.

**Conflicts of Interest:** The author declares no conflict of interest.

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
