# Peer review of "The Dipole Resonator and Dipole Waveguide Insulator in Dense Liquid Medium"

_acoustics, doi:10.3390/acoustics4020029_

Round 1

Reviewer 1 Report

The author theoretically investigated a dipole waveguide insulator in dense liquid medium, where the buoyant force plays an important role. The theory is a generalization of the author’s earlier work in [10], with the addition of the buoyant force. The occurrence of negative effective density and the waveguide locking effect, as well as the broadened bandwidth are quite interesting. The acoustic metamaterials are expected to find a wide spectrum of applications. I would recommend the publication of the manuscript after the following minor revisions.

  1. I found the discussion on the gaseous media at the end of Section 3 confusing. If both m and V0 are zero or approach zero, how can the dipole resonator still be practical or even physical?
  2. The introduction section is a little bit too long for a research article. Perhaps the author could consider trimming it down a bit? This revision is optional.
  3. The figures are a little bit too sketchy. Nowadays scientific publications tend to use more sophisticated illustrations, which are both clear and artistic. This revision is also optional.
  4. In Lines 184 and 187, the indices “2a)” and “2b)” appear to be incorrect.
  5. Throughout the manuscript, ε is used for two different purposes, such as in Eq. (5) and Line 232.

Reviewer 2 Report

Please see attached pdf!

Reviewer 3 Report

The current manuscript investigate the dipole resonator in the dense liquid medium. It can be accepted after some revisions.

1) Figure 1, the distance between Helmhotlz is not uniform. and also the duct wall. The author need to improve the figure for good quanlity. 
2) Figure 4, what is the caption of x and y axis. The author need to improve the figure, and the curve can be different format, such as solid, dashed, dotted. 
3) The same for figure 5, need to improve the quanlity of the figure. can also use different colors.
4) The author can add some results for the absorption which can easily for the reader to understand its engineering applications. 

Reviewer 4 Report

The author theoretically investigated the effective dynamic density for plane acoustic waves propagating along a waveguide embedded with an array of dipole resonators. The novelty is claimed to be the modified theoretical model of the dipole resonator, which takes into account the buoyant force and suggests widened bandgap with negative effective density (particularly becomes nontrivial for dense fluid). However, the contribution of this work is a little bit incremental, and the results are somewhat confusing. The manuscript is also not very clearly written with some vague descriptions or missing information. In addition, a major gap in the current storyline is that there is no clue about how to construct such a system in practice.

Detailed comments:

  1. Similar physical problems (viz. acoustic waveguide connected with a resonator array) have been extensively investigated previously, no matter in the context of conventional duct sound insulation and acoustic liners, or in the context of acoustic metamaterials. In this regard, the manuscript’s current introduction section is not well-organized to include those most seminal and representative works (or review articles) and should thus be substantially improved. Besides, I got the feeling that the author spent a lot of effort in reviewing Refs. [4]-[15] but failed to explicitly point out the research gaps and the motivation of this manuscript.
  2. The schematics (Figs. 2 and 3) together with the descriptions are not very clear. For instance, how does the dipole resonator (mechanical spring-mass resonator) vibrate and couple with acoustic waves? What do L, R, and L_0 (mentioned in line 173) refer to in Fig. 2? What is the spacing distance between two neighboring resonators? Why does the current description involve no Bloch wave for such a periodic system?
  3. Some undefined or unexplained quantities: (i) W_d in Eq. (4) is for particle velocity or acoustic pressure? (ii) What does the surface density n mean? (iii) What is the difference between ? and rho? (iv) Is beta the reciprocal of bulk modulus of the waveguide medium?
  4. For the case of complex resonance frequency Omega, the effective density, speed of sound, and wavenumber should be complex as well. Why are the results in Figs. 4 and 5 purely real and imaginary, respectively?
  5. In Eq. (23), the effective modulus is assumed to be the same as that of the background medium, which seems quite unusual. Could the author please explain this assumption?
  6. Could the authors please provide a realistic structure corresponding the theoretical model under discussion (just like a waveguide with a side-branch array of Helmholtz resonators)? 

Round 2

Reviewer 4 Report

The author's response is not convincing at all. I don't see any revision that has improved the quality of the manuscript, either. I therefore cannot recommend its publication.

Author Response

Dear reviewer, I did my best to respond to the comments of the review:

  1. The text of the introduction (motivation) has been slightly expanded - lines 115-123.
  2. 2. In section 2, a drawing of the design of a dipole resonator experimentally investigated in [10-14] has been added. This design can be the basis for a realistic structure of a dipole waveguide insulator. The corresponding text is added at the end of the conclusion (lines 286-289).
  3. 3. An explanation regarding the compressibility of the medium (lines 217-220) is added.

Round 3

Reviewer 4 Report

I have no further comments.